# Reduction of αSYN Pathology in a Mouse Model of PD Using a Brain-Penetrating Bispecific Antibody

**DOI:** 10.3390/pharmaceutics14071412

**Published:** 2022-07-05

**Authors:** Sahar Roshanbin, Ulrika Julku, Mengfei Xiong, Jonas Eriksson, Eliezer Masliah, Greta Hultqvist, Joakim Bergström, Martin Ingelsson, Stina Syvänen, Dag Sehlin

**Affiliations:** 1Department of Public Health and Caring Sciences, Uppsala University, 751 85 Uppsala, Sweden; ulrika.julku@pubcare.uu.se (U.J.); mengfei.xiong@pubcare.uu.se (M.X.); joakim.bergstrom@pubcare.uu.se (J.B.); martin.ingelsson@pubcare.uu.se (M.I.); stina.syvanen@pubcare.uu.se (S.S.); 2Department of Medicinal Chemistry, Uppsala University, 751 23 Uppsala, Sweden; jonas.p.eriksson@akademiska.se; 3PET Centre, Uppsala University Hospital, 751 85 Uppsala, Sweden; 4Division of Neuroscience and Laboratory of Neurogenetics, NIA-NIH, Bethesda, MD 20814, USA; eliezer.masliah@nih.gov; 5Department of Pharmacy, Uppsala University, 752 37 Uppsala, Sweden; greta.hultqvist@farmaci.uu.se; 6Krembil Brain Institute, University Health Network, Toronto, ON M5T 1M8, Canada; 7Department of Medicine and Tanz Centre for Research in Neurodegenerative Diseases, University of Toronto, Toronto, ON M5T 1M8, Canada

**Keywords:** bispecific antibody, blood-brain barrier (BBB), alpha-synuclein (αSYN), Parkinson’s disease (PD), immunotherapy, monoclonal antibody, transferrin receptor (TfR), receptor-mediated transcytosis (RMT)

## Abstract

Immunotherapy targeting aggregated alpha-synuclein (αSYN) is a promising approach for the treatment of Parkinson’s disease. However, brain penetration of antibodies is hampered by their large size. Here, RmAbSynO2-scFv8D3, a modified bispecific antibody that targets aggregated αSYN and binds to the transferrin receptor for facilitated brain uptake, was investigated to treat αSYN pathology in transgenic mice. Ex vivo analyses of the blood and brain distribution of RmAbSynO2-scFv8D3 and the unmodified variant RmAbSynO2, as well as in vivo analyses with microdialysis and PET, confirmed fast and efficient brain uptake of the bispecific format. In addition, intravenous administration was shown to be superior to intraperitoneal injections in terms of brain uptake and distribution. Next, aged female αSYN transgenic mice (L61) were administered either RmAbSynO2-scFv8D3, RmAbSynO2, or PBS intravenously three times over five days. Levels of TBS-T soluble aggregated αSYN in the brain following treatment with RmAbSynO2-scFv8D3 were decreased in the cortex and midbrain compared to RmAbSynO2 or PBS controls. Taken together, our results indicate that facilitated brain uptake of αSYN antibodies can improve treatment of αSYN pathology.

## 1. Introduction

The number of patients suffering from neurodegenerative diseases will increase due to an aging population in combination with a lack of efficient disease-modifying treatments available for brain disorders. Immunotherapy directed against intrabrain targets remains both a promising and challenging approach to treating neurodegeneration. Parkinson’s disease (PD) is a progressive, degenerative neurological disorder characterized by the presence of pathological, aggregated forms of the protein alpha-synuclein (αSYN), eventually depositing as insoluble, intracellular Lewy bodies (LB), and ultimately, neuronal death. Since the discovery of PD-causing point mutations in the SNCA gene encoding αSYN [1,2] in combination with αSYN being identified as one of the main components of LBs [3], a large body of evidence points toward αSYN as a key target for therapeutic interventions.

However, current treatment options for PD are limited to symptomatic treatment, such as restoration and correction of dopaminergic and cholinergic deficits [4,5]. A number of preclinical studies have demonstrated the effectiveness of both active and passive immunotherapy aimed at the pathological forms of αSYN for halting the disease progression [6,7,8,9]. Despite the development of clinical therapeutic antibodies targeting αSYN thus far showing good safety and tolerability [10,11], the efficacy of immunotherapies in meeting their clinical endpoints has been limited. The underlying reasons behind these results are, in all probability, multifactorial. One important aspect to consider is the form of αSYN to target. Aggregated αSYN has been shown to exhibit acquired toxicity as well as to cause a potential loss of function of normal αSYN [12,13] by altering neurophysiological functions, including impairment of macroautophagy [14] and dysfunction of mitochondria and endoplasmic reticulum [15,16,17,18]. Therefore, conformation selective antibodies that preferentially bind to pathological αSYN over the monomeric, physiologically relevant form are probably necessary for successful αSYN immunotherapy. We have previously demonstrated the reduction of αSYN aggregates in the central nervous system (CNS) of (Thy-1)-h[A30P] transgenic mice following systemic treatment with the murine aggregate-selective antibody mAb47 [7]. In addition, as monomeric αSYN is abundant in blood, it is likely that therapeutic antibodies that bind αSYN monomers will be sequestered in blood and thereby be prevented from engaging with pathological protein forms in the brain. Another major obstacle is the access of the antibody to the pathological, intrabrain αSYN. Monoclonal antibodies are large molecules with restricted brain distribution. One of the main challenges lies within the physical, biochemical, and immune barriers of the CNS comprised of the blood-brain barrier (BBB). The architecture of the BBB is highly dynamic and complex, consisting of specialized brain microvascular endothelial cells (BMECs). The BMECs form tight junctions that regulate paracellular transport, whereas transcellular transport is tightly regulated by transport systems consisting of active transporters, pumps, and receptors [19,20,21]. Restricted direct access to pathology as a major obstacle to treatment efficacy in brain disorders is further strengthened by the fact that many peripheral diseases have been subject to successful immunotherapy [22,23].

Despite the restrictive nature of the BBB, it is the largest brain-blood interface and thus a highly relevant transport route for drug delivery as it allows access to the entire brain volume, which is advantageous when targeting whole-brain disorders such as PD. As the main purpose of the BBB is to maintain the microenvironment in the brain, a non-invasive approach to circumvent the BBB is preferred. One such approach is represented by receptor-mediated transport (RMT). Earlier studies with αSYN targeting single-chain antibodies fused with an LDL receptor-binding domain for increased brain penetration have shown reduced αSYN pathology and amelioration of functional deficits in the Line D and L61 αSYN pathology models [24,25]. The most studied transporter for this purpose is the transferrin receptor (TfR) which is highly expressed in the endothelial cells of the BBB and is involved in the transport of transferrin-bound iron into the brain [26,27,28]. Antibodies can be engineered into bispecific formats, with an additional binding moiety targeting the TfR to enable RMT. Active transport across the BBB will enhance both the amount and the spatial distribution of antibodies, likely expanding the therapeutic potential of brain immunotherapy [29,30,31]. Therefore, active transport is of particular importance when targeting intrabrain αSYN, where the pool of accessible pathology is scarcer in comparison with other proteinopathies such as Alzheimer’s disease (AD).

We have previously expressed several αSYN antibodies in a bispecific format with a single chain variable fragment of the TfR1 antibody 8D3 [32] targeting the TfR as well as aggregated forms of αSYN, resulting in a dramatically increased brain uptake for in vivo αSYN detection in αSYN deposition model mice [33]. Here, we have used one of these antibodies, RmAbSynO2-scFv8D3, for targeting the extracellular pool of aggregated αSYN in a short-term treatment study in the L61 αSYN mouse model, following the characterization of biochemical and pharmacokinetic properties of the antibody.

## 2. Materials and Methods

### 2.1. Animals

Transgenic L61 mice, overexpressing wild-type (WT) human αSYN under the Thy-1 promoter [34,35,36], were bred in-house by crossing B6D2F1 (C57Bl6/DBA2) with heterozygous L61 females on a B6D2F1 background. L61 mice display an age-dependent increase in αSYN pathology, with high levels of oligomeric, aggregated αSYN detected already at 3 months of age [37]. Due to random X-inactivation, insertion of the transgene on the X-chromosome manifests as notable differences in pathology levels and phenotype between males and females. For the pharmacokinetic and microdialysis studies of intravenous (i.v.) and intraperitoneal (i.p.) injections, as well as in the positron emission tomography (PET) study, female WT B6D2F1 mice were used, *n* = 4 for each time point and antibody (*n* = 2 for each administration route), *n* = 2 for PET. For the immunotherapy study, female L61 mice were used, with *n* = 8 for each treatment group. All mice were 14–16 months of age and randomized across treatments by litter and date of birth. Animals were housed in groups of a maximum of five mice in open cages on a 12:12 h constant light cycle in a temperature- and humidity-controlled room and were given food and water ad libitum. All experiments were approved by the Uppsala Animal Ethics Committee (approval 5.8.18-13350/2017, 5.8.18-12230/2019, and 5.8.18-20401/2020) and the use of mice was conducted in accordance with the EU directive 2010/63/EU for animal experiments.

### 2.2. Antibody Design and Expression

The antibodies were based on RmAbSynO2 [38], with a high affinity for large αSYN oligomers and fibrils. RmAbSynO2-scFv8D3 targeting αSYN and mTfR was generated by the fusion of a single-chain variable fragment of the mTfR binding antibody 8D3 [32] to the C-terminal end of the IgG light chain with a short linker as previously described [33,39]. An IgG2C backbone was chosen to stimulate interaction with microglia. In short, the bispecific antibody was recombinantly generated in-house according to a previously published protocol [33] by transient transfection of Expi293 cells with pcDNA3.4 vectors carrying the sequence of either the heavy or the light chain of the antibody with polyethyleneimine as the transfection reagent and valproic acid as the cell cycle inhibitor. The unmodified RmAbSynO2 IgG was generated in a similar way for comparison. For further details on the antibody binding characteristics, see Appendix A.

### 2.3. Radiolabeling

The antibodies were labeled with iodine-125 (^125^I) for ex vivo measurement of the brain and blood distribution in the pharmacokinetics, microdialysis, and treatment studies. Labeling was performed using direct iodination with Chloramine-T [40], as previously described [33]. Molar activity after labeling was 14.6 ± 2.8 MBq/nmol for [^125^I]RmAbSynO2-scFv8D3 and 12.9 ± 2.1 MBq/nmol for [^125^I]RmAbSynO2. The radiolabeling was always performed less than 2 h before the experiments. For PET imaging of brain distribution, antibodies were labeled with fluorine-18 (^18^F), using the inverse electron demand Diels-Alder (IEDDA) click reaction, as previously described [41]. In short, antibodies were chemically functionalized with transcyclooctene (TCO) and then reacted with an ^18^F-labeled tetrazine in PBS. Molar activity after labeling was 116 MBq/nmol for [^18^F]RmAbSynO2-scFv8D3 and 96 MBq/nmol for [^18^F]RmAbSynO2. As a quality control, radiolabeled antibodies were analyzed for binding to αSYN and TfR with ELISA in comparison with non-labeled antibodies, as previously described [33].

### 2.4. In Vivo Experiments and Ex Vivo Measurements

For the pharmacokinetic studies, mice were injected with [^125^I]RmAbSynO2-scFv8D3 (1 mg/kg, 0.35 ± 0.08 MBq) or [^125^I]RmAbSynO2 (1 mg/kg, 0.39 ± 0.09 MBq), either i.p. (*n* = 2/time point) or i.v. (*n* = 2/time point) and sacrificed at 30 min, 2 h, 4 h, or 24 h post-injection. Tail-vein blood samples (8 µL) were obtained at 30 min and 1 h post-injection from the 2 h and 4 h animals, with an additional sample at 2 h from the 4 h animals. From the 24 h animals, blood samples were obtained 1 h, 4 h, and 6 h post-injection. All mice were subject to a terminal blood sample at the time of euthanasia prior to transcardial perfusion with 0.9% NaCl, followed by isolation, subdissection, and flash-freezing of the brain on dry ice. Radioactivity in the brain and blood were measured using a γ-counter (1480 Wizard™; Wallac Oy, Turku, Finland) and expressed as a percentage injected dose/g brain tissue normalized to body weight (%ID/g brain/BW and %ID/g blood/BW). Antibody exposure in blood was expressed area under the curve (AUC_blood_).

For the treatment study, L61 mice were i.v. injected with three doses of 10 mg/kg of either RmAbSynO2-scFv8D3 or RmAbSynO2 (*n* = 8/group). Each dose was supplemented with the radiolabeled antibody of the same type, corresponding to 0.05 mg/kg, to be able to follow the treatment dose pharmacokinetics throughout the study, 0.24 ± 0.09 MBq and 0.29 ± 0.065 MBq of [^125^I]RmAbSynO2-scFv8D3 and [^125^I]RmAbSynO2 respectively. Antibody injections were performed on day 1, 2, and 4, followed by euthanasia by transcardial perfusion with 0.9% NaCl on day 5, 24 h after the last injection. Tail-vein blood samples (8 µL) were obtained 1 h, 4 h, and 24 h after each injection, and a terminal blood sample was collected prior to euthanasia. Brain isolation and radioactivity measurements were performed as above.

### 2.5. Ex Vivo Autoradiography

Frozen 20 µm sagittal sections were prepared with a cryostat (CM1850, Leica Biosystems, Mölndal, Sweden) and mounted on Superfrost Plus glass slides (Menzel Gläser, Braunschweig, Germany). Dried sections were exposed to a phosphor screen (MS, MultiSensitive, PerkinElmer, Downers Grove, IL, USA) for seven days and scanned with Typhoon PhosphorImager (GE Healthcare). The radioactive signal was converted to a false-color scale (Royal) in ImageJ (NIH, Bethesda, MD, USA).

### 2.6. Microdialysis

A guide cannula (AT12.8.iC, AgnTho’s, Lidingö, Sweden) was inserted into the left striatum (coordinates A/P + 0.6, M/L + 1.8 from bregma, and D/V − 2.7 from dura) under isoflurane anesthesia (induction 4% and maintenance 2%; Isofluran Baxter, Baxter S.A., Lessines, Belgium) with stereotaxic surgery. The cannula was attached to the skull with two anchor screws (1 × 2 mm, AgnTho’s) and dental cement (Dentalon plus, Heraeus Kulzer GmbH, Hanau, Germany). Buprenorphine (Bupaq vet, Richter Pharma AG, Wels, Austria) and carprofen (Norocarp vet 50 mg/mL, Norbrook Laboratories Ltd., Newry, UK) were administered subcutaneously for post-operative pain and lidocaine (Xylocaine, Aspen Pharma Trading Ltd., Dublin, Ireland) were used as a local anesthetic. Mice were allowed to recover for 24–48 h before the microdialysis.

Prior to the microdialysis, fluorinated ethylene propylene inlet tubing (FEP PTFE tubing, ID 0.12 mm, AgnTho´s), outlet tubing (Tygon LMT-55, ID 0.13 mm, Ismatec, Cole-Parmer GmbH, Wertheim, Germany), and the probe (AT12.8.1, 1 mm PE membrane, 3 MDa cut-off, AgnTho´s) were coated with 5% polyethyleneimine (PEI MW ~2000, Sigma Aldrich, Saint Louis, MO, USA) 0.5 μL/min for 16 h, and then washed with water 10 μL/min for 10 min following 1 μL/min for 8 h to prevent binding of the radiolabeled antibody in the tubing and probe and to improve probe recovery during the microdialysis as described in [42].

The probe was connected to a push-and-pull microdialysis system containing a CMA 402 Microdialysis syringe pump (CMA Microdialysis AB) and a Reglo ICC Digital Peristaltic pump (CMA Microdialysis AB) and perfused at 0.5 μL/min with Ringer solution containing 0.15% BSA. The mice were quickly anesthetized with isoflurane and injected i.p. or i.v. with [^125^I]RmAbSynO2-scFv8D3 (1 mg/kg; 4.97 ± 0.61 MBq). The probe was inserted into the brain right after the antibody injection. Mice were kept under isoflurane anesthesia (1.2–2%) on a heating pad during the microdialysis. The dialysate was collected 8 × 30 min post-injection. One mouse from each group died after 2–2.5 h after starting the microdialysis; thus, samples were collected only until the time of death from these two mice. The volume of the dialysate was measured by weighing the dialysate, and the peristaltic pump flow rate was adjusted if needed to reach fluid recovery of 97–103%, as described in [43]. Mice were perfused transcardially with saline after the microdialysis and the brain was dissected for the biodistribution measurement. Radioactivity in the dialysate was measured using a γ-counter (1480 Wizard™; Wallac Oy).

### 2.7. PET Imaging

For PET imaging, 14-month-old WT mice (*n* = 2) were anesthetized with sevoflurane, placed in a preheated bed in the PET scanner (Mediso nanoScan system) and injected with a bolus i.v. injection of 5.8 MBq [^18^F]RmAbSynO2-scFv8D3 or 5.9 MBq [^18^F]RmAbSynO2 at the start of PET acquisition. Mice were kept under mild anesthesia and scanned for 120 min with a field of view of 9.8 cm, followed by a CT examination for 5 min. Dynamic PET data were obtained by reconstruction with a Tera-Tomo 3-dimensional algorithm (Mediso) with 4 iterations and 6 subsets. CT raw files were reconstructed using filtered back projection (FBP). All subsequent processing of the PET and CT images was performed with imaging software Amide 1.0.4 [44].

### 2.8. αSYN Extraction

Sequential extraction of αSYN from brain tissue was performed as previously described [37]. In short, flash-frozen brain tissue samples were homogenized with a PreCellys Evolution (VWR, Stockholm, Sweden) in ice-cold Tris-buffered saline (TBS) supplemented with cOmplete protease inhibitor (Roche, Mannheim, Germany) and PhosStop phosphatase inhibitor (Sigma, Gillingham, UK) at a 1:10 *w*/*v* ratio. Next, αSYN species of decreasing solubility were extracted with TBS with 0.5% Triton X-100 (TBS-T) at 16,000× *g* and 70% formic acid (FA) at 100,000× *g*.

### 2.9. ELISA

Levels of different αSYN species and soluble TREM2 (sTREM2) were measured as previously described [37,45] with sandwich enzyme-linked immunosorbent assay (ELISA), using antibodies and enzyme conjugates as summarized in Table 1. For all αSYN ELISAs, half-area 96-well plates were coated overnight with coating antibody at 4 °C. The following day, plates were decanted and blocked for 2 h with 1% BSA at room temperature (RT) and then incubated with brain extracts diluted in 0.1% BSA/0.05 Tween-20 overnight at 4 °C. Plates were incubated with the detection antibody for 1 h, followed by the secondary antibody for 1 h. The sTREM2 ELISA was performed in a similar manner as described above, except that the incubation time for blocking and detecting antibodies was 2 h at RT. For all ELISAs, signals were developed with K-blue Aqueous TMS substrate (Neogen Corporation, Lansing, MI, USA), neutralized with 1 M sulfuric acid, and read at 450 nm on Tecan Infinite Pro (Tecan Group Ltd., Männedorff, Switzerland).

### 2.10. Immunohistochemistry

Immunohistochemistry was performed as previously described [37], with the exception that fresh-frozen cryosections were used. In short, 20 µm sections were fixed with 4% formaldehyde, followed by heat-induced antigen retrieval in 25 mM citrate buffer (pH 6.0) prior to permeabilization with 0.4% PBS-Triton X-100 for 20 min and blocking in 5% normal goat serum for 1 h in RT. Next, sections were incubated overnight at 4 °C with anti-α-syn phospho-Ser129 EP1536Y (pSer129, 1:250, Abcam, Cambridge, UK, ab51253) for detection of pathological αSYN phosphorylated on serine 129 and inflammation markers anti-Iba1 (1:500, WAKO Chemicals, Richmond, VA, USA, 019-19,741) or anti-TREM2 (1:50, R&D Systems, AF1729). The next day, sections were incubated with biotinylated goat anti-rabbit IgG (H + L, 1:250, Vector) for detection of pSer129 and Iba1 or biotinylated goat anti-sheep IgG (H + L, 1:250, Vector) for 30 min, followed by a 30 min incubation with horseradish peroxidase-conjugated streptavidin (1:500, Mabtech AB, Stockholm, Sweden, 3310-9-1000). Sections were developed with NovaRED Chromogen Peroxidase Substrate kit (Vector, SK-4800) according to the manufacturer’s instructions and counterstained with hematoxylin. Lastly, stained sections were washed thoroughly in dH2O, dehydrated and fixed in an ethanol gradient (70–100%) and xylene, mounted with Pertex (Histolab, Gothenburg, Sweden), and left to dry overnight. Sections were washed with PBS between each step except for between blocking and incubation in the primary antibody.

Images were acquired at 4× and 60× magnification with Nikon Eclipse 80i microscope and NIS-Elements BR 4. 20.00 software.

### 2.11. Statistical Analyses

All statistical calculations and analyses were made in Prism 8 (GraphPad Software, Inc. La Jolla, CA, USA). Results are presented as mean ± standard deviation (SD) unless otherwise stated. The levels of [^125^I]RmAbSynO2-scFv8D3 in interstitial fluid (ISF) dialysate were analyzed with the mixed-effects model (REML) for repeated measurements. Brain levels of antibodies upon the termination of the therapy study were analyzed by an unpaired t-test. Brain levels of αSYN and TREM2 were analyzed with a one-way analysis of variance (ANOVA) followed by Sidak’s multiple comparison test.

## 3. Results

### 3.1. Increased Brain Uptake following Intravenous Injections of RmAbSynO2-scv8D

To investigate the binding properties of the recombinantly expressed RmAbSynO2-scFv8D3 and RmAbSynO2 (Figure 1A), an inhibition ELISA was performed, indicating a high selectivity for αSYN aggregates (oligomers and fibrils) over monomers and a high degree of similarity between the two antibody formats (Appendix A). In addition, the affinity to αSYN oligomers and mTfR was not altered after radiolabeling, as validated with ELISA (Appendix A). Antibody administration can be performed through different routes. While intraperitoneal (i.p.) administration is easier to perform, intravenous (i.v.) injections may result in better bioavailability. To investigate the optimal delivery route for a treatment study, the brain and blood pharmacokinetics of [^125^I]RmAbSynO2-scFv8D3 and [^125^I]RmAbSynO2 were evaluated after 1 mg/kg i.v. and i.p. injections in WT mice.

The blood pharmacokinetic profiles of the bispecific and the unmodified antibody displayed similar differences between administration routes. While i.v. administration was characterized by high antibody blood concentration immediately after injection, followed by a gradual decline, i.p. administration was initially low with the highest antibody concentration 4–6 h after injection, followed by a gradual decline. As expected, due to its interaction with peripheral TfR, the overall exposure for RmAbSynO2-scFv8D3 (Figure 1B) was lower compared to RmAbSynO2 (Figure 1C). The AUC_blood_ of RmAbSynO2-scFv8D3 was 3393 and 3036 for i.v. and i.p., respectively, while the AUC_blood_ of RmAbSynO2 was 13,823 and 10,840 for i.v. and i.p. administration, respectively. When comparing brain concentrations following i.v. and i.p. administration of the two antibodies, it was evident that i.p. administration reduced brain concentrations compared to i.v. injections at early time points, i.e., at 30 min, 2 h, and 4 h post-injection (Figure 1D,E). Thus, i.v. injection resulted in substantially higher total brain exposure. At the late time point, 24 h post-injection, brain concentrations were similar after i.p. and i.v. administration. At all studied time points, levels of RmAbSynO2-scFv8D3 in the brain, including brain vasculature, were much higher than for RmAbSynO2. The difference between the mono- and bispecific antibody in total brain concentrations after i.v. administration was 50-fold at the 2 h time point. For both antibodies, the total brain uptake for each administration route was similar across different brain regions (Appendix A).

Ex vivo autoradiography on tissue sections prepared from brains isolated at different time points post-injection confirmed the impact of the administration route on brain delivery with higher signals on sections obtained from i.v. injected animals (Figure 1F). Ex vivo measurements and autoradiography give information on the total concentration of antibodies associated with the brain tissue, including vasculature. To obtain further information about the concentration of free antibodies in the extracellular space of the brain, interstitial fluid (ISF) levels of antibodies were measured by microdialysis over a period of 4 h (Figure 1G). This analysis demonstrated that RmAbSynO2-scFv8D3 ISF concentration in the striatum was significantly higher following i.v. injections in comparison with i.p. (*p* = 0.0313). Overall, the i.v. route was determined to be most efficient for brain delivery of RmAbSynO2-scFv8D3 and was therefore used in the subsequent experiments.

For further in vivo evaluation of whole-brain uptake and distribution of the two different antibody formats after i.v. administration, RmAbSynO2-scFv8D3 and RmAbSynO2 were ^18^F-labeled and visualized with PET imaging in WT mice. During the first 5 min after administration, both antibodies were present at relatively low concentrations in the brain, as compared to surrounding tissues. At 90–120 min post-injection, brain concentration of [^18^F]RmAbSynO2-scFv8D3 had increased, while [^18^F]RmAbSynO2 brain concentration had decreased (Figure 2A), seemingly at a similar rate as the concentration in blood indicated that the signal in the brain region largely originated from the blood volume in the brain rather than from the actual brain tissue (Figure 2B). Similar to its faster elimination from blood, [^18^F]RmAbSynO2-scFv8D3 was low in surrounding tissue.

### 3.2. Increased Treatment Efficacy of RmAbSynO2-scFv8D3 Compared to Its Unmodified Variant RmAbSynO2

In a short-term treatment study in female L61 mice, aged mice received three i.v. injections of 10 mg/kg RmAbSynO2-scFv8D3 or RmAbSynO2 or PBS as a control during one week (Figure 3A). This dosing regime was based on the outcome of the brain and blood pharmacokinetics of RmAbSynO2-scFv8D3 and RmAbSynO2. The blood pharmacokinetics, tracked by trace amounts of ^125^I-labeled antibody mixed in with each treatment dose, showed similar concentrations in blood following the three injections at 1 h, 4 h, and 24 h post-injection (Figure 3B). Despite the low concentration of [^125^I]RmAbSynO2-scFv8D3 in the blood, antibody measurements in the brain at study termination showed significantly higher brain tissue concentration of the bispecific antibody compared to the unmodified antibody (Figure 3C) as well as a significantly higher brain-to-blood ratio (Figure 3D) 24 h after the final dose. In accordance with its low blood concentrations, [^125^I]RmAbSynO2-scFv8D3 was lower than [^125^I]RmAbSynO2 in all peripheral organs except bone, probably due to interaction with TfR in the bone marrow (Appendix A). Taken together, this indicates that [^125^I]RmAbSynO2-scFv8D3 was more efficiently transported into the brain compared to the unmodified format also after repeated injections.

Next, we quantified αSYN levels in brain homogenates from different regions with two separate sandwich ELISAs, measuring the amount of total αSYN and oligomeric αSYN, respectively. While total αSYN levels in TBS soluble brain extracts did not differ between the treatment groups, RmAbSynO2-scFv8D3 treated mice displayed decreased total αSYN in cortical TBS-T extract in comparison with the PBS treated group (Figure 4A).

Measurement of αSYN oligomers, the target of the treatment antibody, indicated increased levels of oligomeric αSYN in TBS extracts from both cortex and midbrain of RmAbSynO2-scFv8D3 treated animals when compared to PBS treated controls. In contrast, αSYN oligomer levels in the TBS-T soluble fraction were decreased in RmAbSynO2-scFv8D3 treated mice, in both cortex and midbrain, compared to the PBS group (Figure 4B). No differences in total αSYN levels were observed in the FA fraction, which represents insoluble αSYN aggregates (Appendix A). Likewise, immunohistochemical staining of pSer129, as a measure of pathological αSYN in different brain regions, did not reveal any differences between the different treatments in comparison with the control group (Figure 4C).

### 3.3. Increased Microglial Response following Antibody Treatment Independent of Format

To investigate the microglial response to treatment with RmAbSynO2-scFv8D3 and RmAbSynO2, levels of soluble TREM2 (sTREM2), a microglial marker, were measured in brain homogenates from treated mice. Compared to PBS injected control mice, levels of sTREM2 were increased after antibody treatment in the TBS fraction of all examined brain regions, independent of the antibody format (Figure 5A). Increased sTREM2 levels were also observed in the TBS-T fraction in the cortex and cerebellum for both antibody formats and in the midbrain for RmAbSynO2-scFv8D3 compared to the PBS-treated animals. Furthermore, increased microglial immunostaining in treated animals was noted on brain sections stained for Iba1, a marker for activated microglia (Figure 5B), as well as for TREM2 (Appendix A). Taken together, this data indicates an increased microglial response to antibody treatment, regardless of the format of the treatment antibody.

## 4. Discussion

Passive immunotherapy by targeting αSYN remains one of the most promising strategies in halting the underlying neuropathological processes in synucleinopathies. Here, we have investigated the brain distribution and therapeutic effects of a brain-penetrating, a bispecific variant of RmAbSynO2, a conformation selective monoclonal antibody targeting aggregated forms of αSYN [38,46]. Several lines of evidence point toward secreted extracellular αSYN as the cause of the spreading of pathology in synucleinopathy brains [47,48,49,50] by seeding the aggregation of endogenous αSYN in recipient cells. However, accessibility to parenchymal pathology of conventional antibodies is severely limited by their restricted passage across the BBB.

We have previously demonstrated that a bispecific form of RmAbSynO2, which in addition to αSYN, also targets the mTfR, is actively transported through the BBB of the entire brain capillary network [33]. As confirmed in the present study, this format dramatically increases antibody brain concentration and mediates its distribution into the whole volume of the brain. We further show that PET scanning can dynamically visualize this process in vivo, showing a rapid and gradual increase in whole-brain uptake of [^18^F]RmAbSynO2-scFv8D3. Importantly, in parallel with PET, antibody entry into the brain ISF was verified with microdialysis. A high ISF concentration of therapeutic antibody is particularly important for the treatment of synucleinopathies, where the protein pathology is mainly intracellular and only low amounts of extracellular pathological αSYN are accessible to be targeted by therapeutic antibodies. Conventional antibodies have been hypothesized to enter the brain via perivascular routes, resulting in slow transport and a low degree of diffusion into the brain parenchyma [51], which could explain why αSYN-directed treatment, in particular, has so far shown a low rate of success. Although antibody brain entry is also limited in AD immunotherapy, Aβ deposits are extracellular and abundant and can therefore be accessed by low antibody concentrations resulting from a slow, gradual leakage into the brain. Indeed, although the clinical outcome is debated, Aβ immunotherapy has demonstrated a robust reduction in plaque load by amyloid PET [52].

Additionally, we demonstrate that the administration route greatly affects brain distribution, especially for the bispecific antibody. RmAbSynO2-scFv8D3 administered i.v. results in more than double the brain concentrations at early time points compared to the i.p. route. This was also seen with ex vivo autoradiography, with a higher radioactive signal in the brain following i.v. injections of RmAbSynO2-scFv8D3. The large differences between administration routes were also evident in ISF concentration of RmAbSynO2-scFv8D3 when measured with microdialysis in the striatum of WT mice, confirming that not only association with brain vasculature but also parenchymal entry is affected. The peritoneal cavity is vast with a large blood supply, in addition to being associated with low impact of stress and thus more suitable for repeated dosing. However, the limited bioavailability offered by this route in comparison with the i.v. route suggests that a high initial systemic concentration is required for optimal TfR-mediated brain delivery.

Here we aimed to study the levels of soluble, membrane-associated and insoluble αSYN following short-term treatment with three i.v. injections of RmAbSynO2-scFv8D3 during five days, in comparison with the unmodified RmAbSynO2 and PBS controls in aged female L61 mice. Monitoring the antibody blood pharmacokinetics following each injection throughout the study revealed similar blood pharmacokinetics and presumably similar brain uptake of the antibodies after each dose. Upon study termination, the levels of RmAbSynO2-scFv8D3 in the brain were higher in comparison with RmAbSynO2, as well as higher in the brain relative to the concentrations in blood. This suggests that repeated dosing is feasible for TfR-mediated brain delivery. In addition, it may indicate target engagement of RmAbSynO2-scFv8D3 with intrabrain αSYN. The difference in brain concentrations between the antibodies was also reflected in the therapeutic effect, where only the bispecific antibody variant significantly altered αSYN levels in the brain. Interestingly, the levels of oligomeric, TBS soluble αSYN were increased in the cortex and midbrain after RmAbSynO2-scFv8D3 treatment. In contrast, RmAbSynO2-scFv8D3 treatment reduced αSYN oligomers in the membrane-associated fraction of the same regions. Aggregated αSYN is proposed to exert cellular toxicity by aberrant interactions with biological membranes [53,54,55]. Our findings point toward a shift in the solubility of the oligomeric αSYN toward a more soluble form, likely induced by engagement of the therapeutic antibody with aggregated αSYN associated with membranes. It is important to bear in mind that this is an acute study conducted over a short period of time, with analysis of tissue only 24 h after the last dose. The observed shift in solubility likely reflects a dynamic transition of αSYN through different states of solubility, where a subsequent clearance of the solubilized αSYN may occur later in the process. In a chronic treatment setting, this could eventually also lead to the removal of less soluble forms of αSYN aggregates, which was not observed here. Still, although antibody treatment did not alter levels of total αSYN in the most soluble brain extracts, levels of membrane-associated total αSYN were lowered in the cortex of RmAbSynO2-scFv8D3 treated mice. However, formic acid-soluble αSYN, as well as immunoreactivity of pSer129 αSYN, used as a proxy for pathological αSYN (i.e., less soluble αSYN aggregates) did not readily mirror changes in αSYN load between the different treatments.

Further, we investigated the microglial response in treated mice, as there is a clear link between released, extracellular pathogenic αSYN and positive microglial inflammatory response [56,57,58]. In addition, microglia-mediated phagocytosis of antibody-target complexes is usually suggested as a plausible mechanism for clearance in CNS immunotherapy. The transition to a pro-inflammatory phenotype is key in the promotion of microglial phagocytosis of pathogenic αSYN and it has been suggested that TREM2, which is highly expressed on the surface of microglia in the CNS, participates in this process [59]. In addition, mutations in TREM2 have been identified as risk factors for several neurodegenerative diseases, including PD [60]. Here, sTREM2 levels were significantly increased in all brain regions following both the RmAbSynO2-scFv8D3 as well as RmAbSynO2 treatment as compared with PBS controls. The impact of antibody treatment, regardless of format and brain concentrations, on the TREM2 levels indicates that a common mechanism is at play. Increased immunoreactivity of Iba1, a marker for activated microglia, in addition to TREM2, was also observed on brain sections from both antibody treatment groups, further verifying that treatment induced an increased microglial response. However, to fully understand the impact of the antibody treatment on microglial response, an isotype-matched, non- αSYN-targeting control antibody should be studied in parallel.

## 5. Conclusions

Taken together, this study demonstrates that RMT greatly enhances the brain distribution of a therapeutic antibody that specifically targets toxic αSYN aggregates in a mouse model of αSYN pathology. The rapid and widespread distribution of the antibody into the brain at high concentrations facilitates the clearance of αSYN oligomers.

## Figures and Tables

**Figure 1 pharmaceutics-14-01412-f001:**
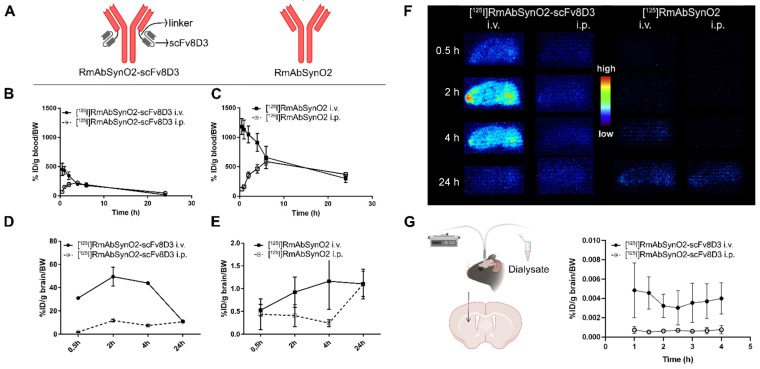
(**A**) Design of the bispecific RmAbSynO2-scFv8D3 with a single chain variable fragment of the mTfR binding antibody 8D3 fused to the C-terminus of the light chain with a short linker and the unmodified RmAbSynO2. (**B**) Blood exposure of [^125^I]RmAbSynO2-scFv8D3 and (**C**) [^125^I]RmAbSynO2 following intravenous (i.v.) and intraperitoneal (i.p.) injections. Blood exposure of both antibody formats was higher after i.v. compared to i.p. administration, with [^125^I]RmAbSynO2-scFv8D3 displaying lower total exposure. (**D**) Total brain concentrations of [^125^I]RmAbSynO2-scFv8D3 and (**E**) [^125^I]RmAbSynO2 were both higher after i.v. injections in comparison with i.p., with [^125^I]RmAbSynO2-scFv8D3 concentrations peaking 2–4 h post-injection and substantially higher than [^125^I]RmAbSynO2 at all points (NB different y-axis scale). (**F**) Radioactive signal in 20 µm sagittal brain cryosections from WT mice in (**D**,**E**), demonstrating that mice i.v. injected with [^125^I]RmAbSynO2-scFv8D3 showed the highest signals at all time points. (**G**) Levels of [^125^I]RmAbSynO2-scFv8D3 were higher in interstitial fluid (ISF) dialysate measured by microdialysis in the striatum after i.v. injections compared to after i.p. injections (mean ± SEM, *p* = 0.0313, *n* = 4/administration route). All blood and brain concentrations are expressed as % injected dose per gram tissue, normalized to bodyweight (%ID/g tissue/BW).

**Figure 2 pharmaceutics-14-01412-f002:**
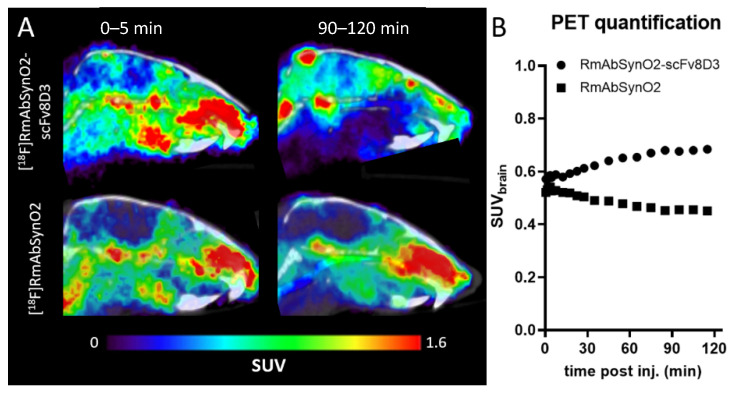
(**A**) Sagittal PET images of WT mice injected with [^18^F]RmAbSynO2-scFv8D3 (upper; *n* = 1) or [^18^F]RmAbSynO2 (lower; *n* = 1) at 0–5 min and 90–120 min post-injection. (**B**) Quantification of brain (whole brain except cerebellum) activity concentration (SUV) from 0–120 min after antibody injection in same animals as (**A**).

**Figure 3 pharmaceutics-14-01412-f003:**
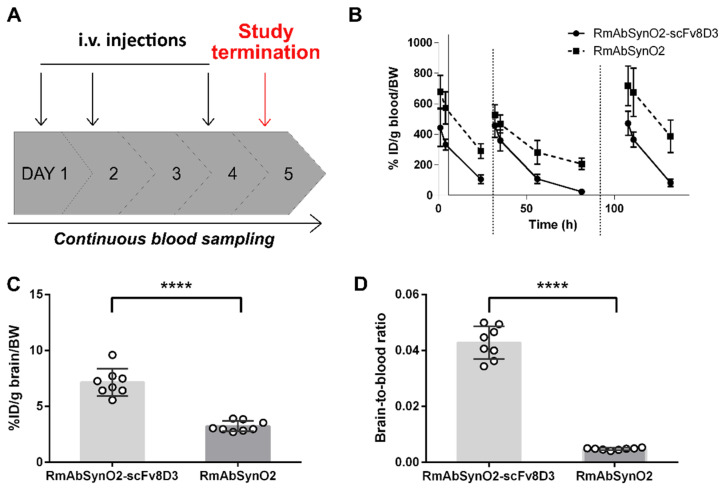
(**A**) Study design of a short-term treatment study in female L61 αSYN mice between 14-16 months of age. Mice were treated with i.v. injections 10 mg/kg RmAbSynO2-scFv8D3 or RmAbSynO2 or PBS (*n* = 8 per treatment group) on day 1, 3 and 4, with study termination on day 5. (**B**) Lower blood exposure of [^125^I]RmAbSynO2-scFv8D3 compared with [^125^I]RmAbSynO2 at 1 h, 4 h and 24 h after each injection. (**C**) Higher concentration of [^125^I]RmAbSynO2-scFv8D3 in the brain compared to [^125^I]RmAbSynO2 (7.1 ± 0.43 and 3.2 ± 0.16%ID/g brain/bw, respectively) upon study termination. (**D**) Higher brain-to-blood ratio of [^125^I]RmAbSynO2-scFv8D3 (0.043 ± 0.0021) in comparison with [^125^I]RmAbSynO2 (0.0048 ± 0.00013). All values are presented as means ± SD and analyzed with an unpaired *t*-test. **** *p* < 0.0001.

**Figure 4 pharmaceutics-14-01412-f004:**
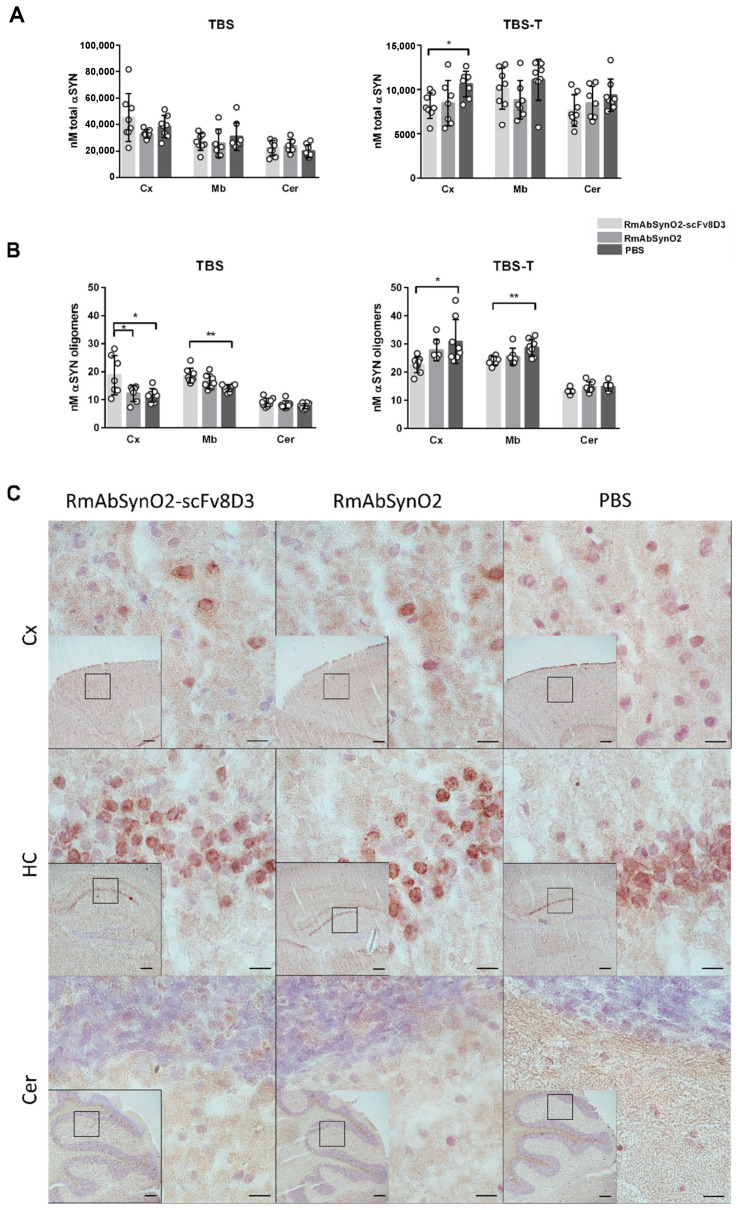
(**A**) Levels of total αSYN in TBS and TBS-T brain extracts measured with MJFR1/Syn-1 enzyme-linked immunosorbent assay (ELISA) measuring all forms of αSYN. No differences in total αSYN in the cortex (Cx), midbrain (Mb), and cerebellum (Cer). Reduced total αSYN in Cx in the TBS-T fraction in mice treated with RmAbSynO2-scFv8D3 in comparison with the PBS group. (**B**) Oligomeric αSYN levels in TBS and TBS-T extracts measured with a homogenous MJFR-14-6-4-2 ELISA. Oligomeric αSYN in the TBS fraction was increased in Cx in the RmAbSynO2-scFv8D3 group in comparison with both the RmAbSynO2- and the PBS control groups and in Mb compared to the PBS group. Oligomeric αSYN levels in the TBS-T fraction were instead decreased in Cx and Mb in the RmAbSynO2-scFv8D3 group compared to PBS controls, while Cer αSYN levels were unchanged. (**C**) Representative images of 20 µm cryosections stained for Iba1 in Cx, hippocampus (HC), and Cer at 60× magnification, with 4× magnifications embedded with squares representing the magnified area. No differences in immunoreactivity were noted between the treatment groups. All values are presented as means ± SD, with a one-way analysis of variance followed by Sidak’s multiple comparison test. * *p* < 0.05, ** *p* < 0.005. Scale bars: 200 µm in embedded images, 10 µm in magnified images.

**Figure 5 pharmaceutics-14-01412-f005:**
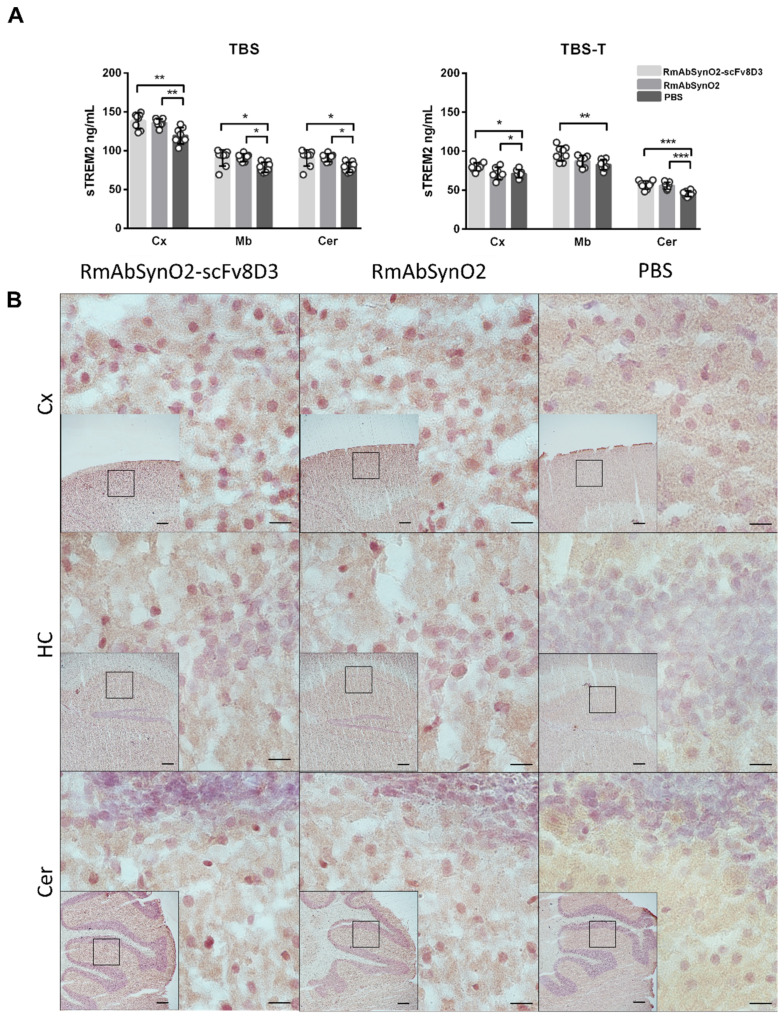
(**A**) ELISA quantification of sTREM2 levels in TBS and TBS-T extracts from the cortex (Cx), midbrain (Mb), and cerebellum (Cer), following treatment with RmAbSynO2-scFv8D3 or RmAbSynO2 in comparison with PBS. Both antibody treatment groups displayed increased sTREM2 levels in both TBS and TBS-T extract of most examined regions. (**B**) Representative images of 20 µm cryosections stained for Iba1 in Cx, hippocampus (HC), and Cer at 60× magnification, with 4× magnifications embedded with squares representing the magnified area, showing increased immunoreactivity on sections from mice treated with both antibody formats. All values are presented as means ± SD, with a one-way analysis of variance followed by Sidak’s multiple comparison test. * *p* < 0.05, ** *p* < 0.005, *** *p* < 0.001. Scale bars: 200 µm in embedded images, 10 µm in magnified images.

**Table 1 pharmaceutics-14-01412-t001:** The concentration of antibodies and enzyme conjugates used in sandwich ELISAs.

Assay	Oligomeric αSYN	Total αSYN	sTREM2
**Coat**	MJFR-14-6-4-2 (Abcam, ab209538) 1 μg/mL	MJFR1 (Abcam, ab138501) 0.25 μg/mL	AF1729 (R&D, AF1729), 0.5 µg/mL
**Detection**	MJFR-14-6-4-2 biotinylated (Abcam, ab209538) 1 μg/mL	Syn-1 (BD Biosciences, 610787) 0.35 μg/mL	BAF1729 (R&D, BAF1729), 0.5 µg/mL
**Enzyme conjugate**	SA-HRP (Mabtech AB, 3310-9-1000) 1:2000	anti-mouse IgG F (ab’)_2_ (Jackson Immuno Research, 115-036-006) 0.4 μg/mL	SA-HRP (Mabtech AB, 3310-9-1000) 1:2000

## Data Availability

Imaging data is available in Dicom or text format and can be transferred per request by the corresponding author. Processed data, i.e., %ID/g/BW, αSYN concentrations, are available in table format in GraphPad Prism files.

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
