# Peer review of "Reduction of αSYN Pathology in a Mouse Model of PD Using a Brain-Penetrating Bispecific Antibody"

_pharmaceutics, 2022, doi:10.3390/pharmaceutics14071412_

Round 1

Reviewer 1 Report

In this paper, the authors study the therapeutic effect of bi-specific antibody against TfR1 and αSYN in human αSYN expressing mice. Many researchers have used 8D3 (antibody against TfR1) or its fragments as a tool to induce receptor-mediated transcytosis to cross the BBB, although its brain penetrating ability is not so high. The anti-αSYN antibody, the authors developed previously, has a novel function and the therapeutic effect of scFv 8D3-fused antibody evaluated by the solubility of αSYN oligomers in the brain is very interesting and important results. The Materials and Methods section is well-written but my major concerns are poor explanation of the aim of each experiments and lack of an appropriate control through of experiments.  

Major concerns

1.     In Figures 1, why did the authors perform i.p. injection? The aim of this experiment is poorly explained and the data is not well-connected to the other data. If there is a possibility that the competition of TfR by high concentration of the antibody cause low efficiency of brain uptake or something, please mention it. Or do the authors want to show the difference between bolus-injection and continuous infusion of the antibody using mice?

2.     In abstract, ‘In addition, the treatment resulted in a microglial response as indicated by higher levels of sTREM2 and increased Iba1 immunoreactivity.’
      It causes misleading. According to the data, microglial responses in this short-term treatment period do not contribute to changing the solubility of αSYN by 8D3-conjugated antibody because there is no difference in sTREM2 level between conjugated and non-conjugated antibody treatment groups. Moreover, non-targeting control antibody must be used to show whether microglial responses are induced by the injection of antibody targeting αSYN. The isotype matched control antibody should be used.
       And in conclusions, ‘The rapid and widespread distribution of the antibody into the brain at high concentrations facilitates clearance of αSYN oligomers and mediates a microglial response that is likely involved in αSYN clearance.’
It is not supported by the data. 

3.     Page 8 lines 321-323, ‘The difference between the mono- and bispecific antibody in brain concentrations after i.v. administration was 50-fold at the 2 h time point. For both antibodies, the brain uptake for each administration route was similar across different brain regions (Suppl Fig 2)’ 
      This may cause misleading because majority of injected bi-specific antibodies are still trapped into/onto the vascular endothelial cells and are not entered into the brain (https://www.science.org/doi/abs/10.1126/scitranslmed.3002230, https://molpharm.aspetjournals.org/content/80/1/32.full). The brain vascular endothelial cells are not actual brain tissue, like the authors described in lines 343-344. The antibody concentration in brain ISF should be used as the brain concentration.

And Page 13 lines 503-504, ‘Importantly, in parallel with PET, efficient antibody entry into the brain ISF was verified with microdialysis’.
     It is not supported by the data. To claim ‘efficient’ antibody entry, the authors must use mono-specific antibody as a control. 

Minor

1.     According to the previous reports published by the authors (https://www.sciencedirect.com/science/article/pii/S0969996115001564, https://www.sciencedirect.com/science/article/pii/S0028390822000442),  RmAbSynO2 is IgG1. Why did the authors choose an IgG2c backbone to make the fusion antibody? Mouse IgG2c may massively stimulate FcγRs on microglia. If there is a specific reason, please mention it.

Reviewer 2 Report

In this manuscript, the authors used a bispecific antibody to reduce the aSYN in the PD mouse model. The authors have studied the pharmacokinetic properties of the antibody.  The proposed approach would be useful for the treatment of not only PD but also brain diseases. The manuscript can be published after minor revisions:

1) What is the binding affinity of RmABSynO2-scFv8D3 antibody to transferrin receptors?

 2) The authors have demonstrated the accumulation of RmABSynO2-scFv8D3 antibodies in the brain. The authors should show the accumulation of the antibody in other organs after i.v. administration.     

 3) Is intranasal administration more effective than i.v. or i.p. administrations for higher accumulation of the antibody in the brain?

 4) Write the full form of TBS and TBS-T in the manuscript.

 5) The authors should demonstrate the behavioral tests of PD animals after administration of antibodies to validate their therapeutic efficacies.

Author Response

REVIEWER 2

Comments and Suggestions for Authors

In this manuscript, the authors used a bispecific antibody to reduce the aSYN in the PD mouse model. The authors have studied the pharmacokinetic properties of the antibody.  The proposed approach would be useful for the treatment of not only PD but also brain diseases. The manuscript can be published after minor revisions:

1) What is the binding affinity of RmABSynO2-scFv8D3 antibody to transferrin receptors?

Response: This has been investigated in previous reports with antibodies in the exact same format, in which we found the binding affinity to be approximately 0.2 nM. This has been clarified in the supplementary information of manuscript.

 2) The authors have demonstrated the accumulation of RmABSynO2-scFv8D3 antibodies in the brain. The authors should show the accumulation of the antibody in other organs after i.v. administration.    

Response: We agree with the reviewer, and the accumulation of the antibody after acute treatment study termination has been added to the manuscript as supplementary figure 3A.

 3) Is intranasal administration more effective than i.v. or i.p. administrations for higher accumulation of the antibody in the brain?

Response: Intranasal administration is a very interesting approach that we are yet to investigate. However, in this study we want to be able to access the whole volume of the brain by means of receptor mediated transcytosis through the BBB of the brain capillary network. We therefore believe that taking advantage of the entire brain vasculature by administration in the blood is advantageous.

 4) Write the full form of TBS and TBS-T in the manuscript.

Response: The full form of TBS (Tris-buffered saline) and TBS-T (TBS with 0.5% Triton) has been written in the methods section 2.8 (αSYN extraction).

 5) The authors should demonstrate the behavioral tests of PD animals after administration of antibodies to validate their therapeutic efficacies.

Response: We agree with the reviewer that this is a very important aspect of therapeutic outcome that would be interesting to study. However, we believe that it is better suited for long-term therapy studies and thus beyond the scope of the current study.

Round 2

Reviewer 1 Report

From many articles, approximately 0.1%ID/g brain (=0.005%ID/g brain/BW) of IgG can enter the brain after i.v. injection of 1020 mg/kg of IgG into mice. Thus, I think the efficiency of brain entry of RmAbSynO2-scFv8D3 is comparable to that of RmAbSynO2 and suggested to use control antibody if the authors want to claim ‘efficient’ brain entry or brain ‘penetration’ or ‘uptake’. According to Figures 1D and 1G, approximately 10%ID/g brain/BW of IgG was in whole brain and 0.004%ID/g brain/BW of IgG was in ISF, indicating that main delivery site of RmAbSynO2-scFv8D3 is brain vasculature, not brain parenchyma. The authors should change ‘brain-penetrating bispecific antibody’ to ‘anti-aggregated αSYN/TfR1 bispecific antibody’ . To connect the data well, I suggest the authors should discuss the perivascular αSYN deposition. Some reports indicate that αSYN deposition could be found in the perivascular basement membranes. It is very important to clarify where is the targeting site of RmAbSynO2 in the brain sections by staining with vascular (or other cellular) marker and RmAbSynO2 (of course by higher magnification).

In abstract, the authors should use Triton X-100 soluble aggregated αSYN, instead of TBS-T. The authors should amend abstract well after the revision.

In Materials and Methods section, the authors should mention RmAbSynO2 IgG was generated using IgG2c backbone as well.